# Pituitary Apoplexy: Risk Factors and Underlying Molecular Mechanisms

**DOI:** 10.3390/ijms23158721

**Published:** 2022-08-05

**Authors:** Betina Biagetti, Rafael Simò

**Affiliations:** Diabetes and Metabolism Research Unit, Vall d’Hebron University Hospital and Vall d’Hebron Research Institute (VHIR), Universidad Autónoma de Barcelona, 08035 Barcelona, Spain

**Keywords:** apoplexy, molecular, pituitary, VEGF, AIP, TNF, growth factors, hypoxia-inducing factor, matrix metalloproteinase, review

## Abstract

Pituitary apoplexy is a rare syndrome, graded from asymptomatic subclinical apoplexy to a life-threatening condition due to pituitary ischemia or haemorrhage of an enlarged pituitary gland. The risk factors and the molecular underlying mechanisms are yet to be elucidated. We provide an overview of the general concepts, the potential factors associated with pituitary adenoma susceptibility for apoplectic events and the molecular mechanisms that could be involved such as HIF-1α/VEGF pathways and metalloproteinases activation, among others. The knowledge of the molecular mechanisms that could participate in the pathogenesis of pituitary apoplexy is crucial to advancement in the identification of future diagnostic tools and therapeutic targets in this rare but sometimes fatal condition.

## 1. Introduction

Pituitary apoplexy can be a life-threatening condition as a result of hemorrhage and/or infarction of an enlarged pituitary gland by a tumor or non-tumor process [1,2]. In these cases sudden onset headache associated or not with visual disturbances or neurological signs are the main symptoms. However, some cases are subclinical, only revealed by magnetic resonance imaging (MRI) performed during the follow-up of known pituitary adenomas or due to other reasons.

Pituitary apoplexy is bleeding into the pituitary gland. Regarding the etiology, there is some evidence that the growing tumor outstrips its blood supply causing ischemic necrosis. Similarly, in the cases of non-adenomatous pituitary apoplexy (i.e., Sheehan syndrome), the hyperplasia of the pituitary gland during pregnancy increases the demand for blood supply resulting in ischemia [3]. There are several risk factors, such as some drugs (e.g., antiplatelet agents, anticoagulants, dopamine agonists, gonadotrophin agonists), previous surgery, brain trauma, endocrine dynamic tests and systemic diseases [4]. Non-functioning pituitary adenomas (NFPA), specifically macroadenomas, appear to have the higher risk of apoplexy [5,6,7]. Nevertheless, apoplexy in giant pituitary adenoma is extremely rare [8]. The involvement of risk factors such as microvasculopathy [9] or the prothrombotic state (e.g., patients with uncontrolled diabetes [10,11], or pregnancy [3]) has also been postulated. Some data also support that some genetic conditions, cytokines and growth factors might trigger some pathways, leading to necrosis or hemorrhage of the pituitary [12,13]. However, the involved mechanisms are far from being fully understood. This review will give an overview on the risk factors and underlying molecular mechanisms involved in the pathogenesis of pituitary apoplexy.

## 2. Methods

This narrative review about molecular mechanisms in pituitary apoplexy was conducted following the SANRA scale [14]. We have carried out the search in PubMed for articles published in English, with the following strategy:(“apoplexy”[Title/Abstract] AND “molecular”[Title/Abstract]) AND (english[Filter]): 18 articles were found((“apoplexy”[Title]) AND (review[Filter]) AND (english[Filter])): 116 articles found

The 132 articles identified by these searches and relevant references cited in those articles were reviewed. We largely selected those published in the past 20 years but did not exclude seminal older articles.


*Pituitary Development and the Portal System*


The pituitary gland has a dual embryonic origin (neuroectodermal and non-neuroectodermal). The posterior lobe (neurohypophysis) consists of nervous tissue arising from the embryonic forebrain and represents an extension of the hypothalamus.

The anterior lobe (adenohypophysis), is derived from an outpouching of the roof of the pharynx (Rathke’s pouch) [15].

Both the adenohypophysis and neurohypophysis can be subdivided based on gross and histologic features.


Adenohypophysis:
Pars anterior or distalis: this is the largest part and is responsible for hormone secretion.Pars tuberalis or infundibulum: this is an upwards extension of the pars anterior and wraps the pituitary stalk [15].Pars intermedia: this is a thin epithelial layer that separates the pars anterior from the posterior lobe [15].



Neurohypophysis:
Pars nervosa—the portion of the posterior pituitaryMedian eminence—the upper section of the neurohypophysis above the pars tuberalisInfundibular stalk—connects the pars nervosa to the base of the brain


The pituitary gland is a well-vascularized tissue. Hormone-exchanging blood vessels between the hypothalamus and the pituitary gland and the pituitary portal system can be observed in early developmental stages of the fetus. In humans, it has been observed that the hypophyseal portal system is fully developed by week 11.5. This was determined by injecting a silicone rubber compound into specimens of various stages of gestation. In a specimen at week 11.5, the median eminence and infundibular stalk contained the compound, suggesting the existence of the fully developed portal system [16]. This close relationship between vessels and gland development is paramount for the growth of the developing fetus. Ultimately, the adult pituitary receives its blood supply from a vascular network comprised of the superior and inferior hypophyseal arteries and the portal system. The infundibulum, the median eminence and the pars tuberalis are supplied by the superior hypophyseal artery (a branch of the internal carotid artery) and the posterior pituitary receives its blood supply directly from the inferior hypophyseal artery (Figure 1). Conversely, the anterior pituitary is supplied indirectly from two sources. Long portal vessels supplying 70% of the blood to the gland arise from the hypothalamus, which in turn is supplied by the superior hypophyseal artery. Short portal vessels supply 30% of blood to the gland and arise from the posterior pituitary. These portal vessels are exquisitely sensitive to volume and pressure changes in the systemic circulation.

The hypophyseal portal vessels located in the median eminence are fenestrated and constitute one of several neurohaemal junctions. These junctions facilitate the release of chemical messengers from nerve terminals into the bloodstream and vice versa [17].


*Predisposing and Precipitant Factors for Pituitary Apoplexy*



Precipitant factors:


There are some precipitating factors described in the literature and identified in 10–40% of cases as major previous surgery, brain trauma, endocrinological dynamic test, angiographic procedures, bleeding disorders, as well as some drugs (e.g., antiplatelet, anticoagulants, dopamine agonists or gonadotrophin agonists) [18].


Cardiovascular risk factors:


The classic and well-established cerebrovascular and cardiovascular risk factors such as diabetes, hypertension, and dyslipidemia, are not well-established risk factors for pituitary apoplexy.

The presence of diabetic microangiopathy and/or hemodynamic abnormalities and the prothrombotic state that occurs in patients with uncontrolled diabetes could favour pituitary apoplexy [9,10,11]. However, some reports have not found this association in either acute apoplexy [18,19] or subclinical apoplexy [20]. Likewise, in relation to hypertension there are published results in favor [21] and against [18] its participation as a risk factor for pituitary apoplexy.


Genetic markers:


Aryl hydrocarbon receptor-interacting protein (AIP) germline mutations typically lead to young onset macroadenomas with aggressive behavior [22,23]. This is an autosomal dominant and low penetrance mutation present in approximately 20% of the relatives of patients with familial isolated pituitary adenoma (FIPA), and in 8% to 20% of apparently sporadic macroadenomas in patients younger than 30 years [23,24]. AIP mutation has been linked to pituitary apoplexy [12,23]. However, no AIP mutation was found in a large cohort reported by Fialho et al. [25] and, therefore, the identification of AIP mutation in patients who have suffered from pituitary apoplexy seems irrelevant in clinical practice.


Tumor characteristics:


Pituitary apoplexy has been described as more frequent in NFPA [5,6,7], followed by prolactinomas [7,26] and GH tumors [2]. In fact, NFPAs are more frequent and generally larger than functional adenomas and, for this reason; pituitary apoplexy could be more frequent in these types of tumors. In addition, data from hospitalized patients suggest that pituitary apoplexy is less frequent and has a more favorable prognosis in microadenoma [27].

A large surgical series suggested that the clinical picture of pituitary apoplexy could be graded based on clinical presentation from subclinical to severe symptomatic with acute visual deficits or decreased Glasgow scale [28]. Intrinsic factors related to pituitary tumor growth (i.e., tumor volume, cavernous sinus involvement, suprasellar extension) could contribute to make it more prone to bleeding. In close relationship with this concept, McCabe et al. [29] investigated the expression of vascular endothelial growth factor (VEGF), which is closely related to vascularization and tumor growth, in 103 human pituitary tumors, of which (n = 81) were NFPA. They demonstrated markedly raised VEGF mRNA in NFPA.


Hemodynamic conditions:


The unique circulation type in the pituitary gland consisting of a portal system, a relatively low-pressure system, makes the pituitary cells more susceptible to ischemia. Conversely, the posterior pituitary gland, as we explained above, receives its blood supply through the inferior hypophyseal artery which functions under higher pressure than the anterior pituitary. Therefore when the reason of pituitary apoplexy is due to low-pressure or hypovolemia the adenohypophysis is affected as a whole, whereas the neurohypophysis is generally preserved [30].

Sheehan syndrome is a massive pituitary bleeding that occurs postpartum as a result of an ischemia of the pituitary gland due to a significant uterine hemorrhage during childbirth [3]. This is facilitated by compression of the superior hypophyseal vessels against the sellar diaphragm due to the pregnancy-induced enlargement of the pituitary gland.

Hypopituitarism can also appear after traumatic brain injury [31] and in some cases of hypovolaemic shock, among others [32,33,34], as a result of a certain degree of ischemia of the pituitary gland.

On the other hand, increased pressure within an adenoma begins to occur with tumors as small as 1–2 mm, and it is a contributing pathogenic factor of intra-tumor hemorrhage [35].


*Clinical Presentation and Treatment of Pituitary Apoplexy*


The clinical presentation of pituitary apoplexy is widely variable, in part determined by the extent of hemorrhage or necrosis [2]. Severe sudden onset headache is one of the main symptoms with or without visual disturbance, and some neurological signs such us photophobia, nausea or vomiting [1,2]. However, frequently a subclinical pituitary apoplexy is incidentally discovered in the tomography or magnetic resonance imaging (MRI) performed for other conditions.

The frequency of subclinical pituitary apoplexy has been estimated in 10–25% of patients with pituitary adenoma [1,2,7,36], whereas symptomatic acute pituitary apoplexy has been reported less frequently (2–10% of the patients) (Figure 2) [1,7].

Regarding treatment, traditionally acute pituitary apoplexy has universally been considered as a neurosurgical emergency [37]. Currently, the best treatment of pituitary apoplexy is a matter of debate. A conservative approach using high doses of glucocorticoid is mandatory immediately after diagnosis, and neurosurgery is reserved to more severe cases or when the conservative approach has failed [38,39,40,41,42]. However, there is not agreement on the best timing of surgery when there is visual impairment [43,44].


*Pathophysiology of Pituitary Apoplexy*


Little is known regarding pituitary apoplexy mechanisms. We can find two scenarios, the most frequent is the ischemic, hemorrhagic or mixed apoplexy of an adenoma and the second is apoplexy of the whole pituitary gland.

Although the vascularization of pituitary adenomas is supported by a direct arterial blood supply rather than the portal system [2,45]; the intrinsic adenoma vascularity is reduced compared with the normal pituitary gland, as has been reported by counting vessel density [46] and by contrast enhanced imaging [47]. In addition, the adenoma vascularity seems to be more fragile, with signs of incomplete maturation and basal membranes often ruptured [48,49]. On the other hand, the high metabolic and energy requirement needed by the tumor to grow could outgrow their blood supply. In this setting the acute reduction in systemic blood pressure may decrease blood flow to the pituitary adenoma and precipitate apoplexy [2]. Dynamic tests aimed at evaluating pituitary function can acutely increase the metabolic demands of the tumor, thus also precipitating apoplexy. Compression of infundibular or superior hypophyseal vessels against the sellar diaphragm by the expanding tumor mass has also been postulated as a cause of apoplexy. However, the adenoma is supplied by the inferior hypophyseal artery, thus the infundibular compression could lead to adenohypophysis ischemia and hypopituitarism rather than adenoma ischemia.


*Molecular Mechanisms Underlying Pituitary Apoplexy*


The main risk factors and molecular mechanisms involved in the pathogenesis of pituitary apoplexy are displayed in Figure 3.


Vascular endothelial growth factor


Increased angiogenesis is essential for the invasiveness and spread of many types of tumors including pituitary tumors. Conversely, compared with the normal gland, pituitary tumors (which are slow growing tumors) have a reduced vessel density [46,47], a reduced angiogenesis and a high intra-tumor pressure [47].

VEGF, transforming growth factor beta (TGF-β), and Wnt are signalling pathways influencing the vasculature formation.

As in other tissues, VEGF is involved in the angiogenesis of pituitary adenomas [50]. The evidence regarding VEGF levels in pituitary tumors is controversial. Some authors, as mentioned above, found markedly raised levels of VEGF in NFPA [29] while others reported negligible levels as occurs in the normal brain [51]. Lloyd et al. [52] found that VEGF expression was low in adenomas compared with a normal pituitary gland, but in pituitary carcinomas VEGF expression was higher [52]. In agreement with this finding, more recent research has found a link between VEGF and pituitary adenoma invasiveness [53,54]. Moreover, Lee et al. [55] studied the microvascular density of apoplectic pituitary using vascular endothelial markers and only VEGF was found to be positively correlated. Taken together, these findings suggest that an up-regulation of VEGF occurs during pituitary tumor progression [52]. VEGF secreted by tumor cells promotes neovascularization via downstream pathways including the MAPK signaling pathway, FAK pathway, PI3K/Akt pathway, and p38 MAP kinase pathway, directly stimulating tumor cell proliferation [56].


Cyclooxygenases


Cyclooxygenases promote prostaglandin formation that can trigger cellular signalling pathways mediating vasodilation, platelet aggregation, and inflammation among others. Akbari et al. [57] studied the expression level of the isoforms of cyclooxygenase (COX-1 and COX-2) in pituitary tumors. A significant expression level of COX-2 was observed in NFPA compared with other pituitary tumors. Furthermore, the COX-2 expression level was significantly increased in invasive tumors. Additionally, the prostaglandin 2 level was increased in macroadenoma compared with microadenoma and in invasive compared with non-invasive pituitary tumors. Therefore, although there is no specific information on cyclooxygenases in pituitary apoplexy, it seems reasonable to postulate their involvement in the pathogenesis of pituitary apoplexy.


Wnt signaling


Wnt signaling is one of the pivotal evolutionarily conserved networks that orchestrates cell–cell communication, cell proliferation, differentiation and migration during embryonic development [58]. The canonical Wnt pathway (or Wnt/β-catenin pathway) is the Wnt pathway that causes an accumulation of β-catenin in the cytoplasm and its eventual translocation into the nucleus to act as a transcriptional coactivator of transcription factors that belong to the TCF/LEF family. There is some evidence regarding Wnt dysregulation in human and murine pituitary tumors [59]. Reduction of E-cadherin/catenin complex is related to pituitary tumor invasiveness [60]. Crosstalk between integrins and beta-catenin pathways has been suggested in several tumor tissues [61]. Epithelial integrin β1 functions as a regulator of developmental angiogenesis in the pituitary gland, thus providing insight into how vascular invasion is coordinated with tissue development [62].


Tumor Necrosis Factor α (TNF-α) and hypoxia inducible factor 1 (HIF-1α)


TNF-α is a cytokine involved in multiple pathological pathways as inflammation, angiogenesis, vascular hyperpermeability, and destruction of vascular integrity. Xiao et al. [63] found higher expression levels of TNF-α, VEGF, and matrix metallopeptidase-9 in hemorrhagic pituitary adenomas than in non-hemorrhagic ones obtained from human surgical specimens [63]. In addition, the same investigators [64] performed an in vivo study using MMQ (a prolactin-secreting clonal cell line) cell xenografts mice showing that HIF-1α overexpression significantly promoted hemorrhagic transformation [64]. HIF 1-alpha was also linked with an anti-apoptotic role in pituitary adenoma cell lines in hypoxia [65]. Therefore, tumor hypoxia, as a consequence of the rapid tumor growth, may promote an increase of TNF-α and VEGF via HIF-1α signalling pathway as an adaptive process, but it can also be a significant factor contributing to the development of pituitary apoplexy.


Epidermal Growth Factor-Receptor (EGFR)


The epidermal growth factor (EGF), binding through EGF-receptor (EGFR), is a potent modulator of cell proliferation and differentiation. Expression of EGF and EGFR has been detected in both the normal pituitary gland and in pituitary adenomas [66,67]. Kong Y et al. [68] found increased levels of soluble EGFR (sEGFR) in peripheral serum in patients with pituitary adenoma with a relationship with tumor size and the presence of pituitary apoplexy. Although these findings suggest that serum sEGFR might be a marker of proliferation and apoplexy in pituitary adenoma, further confirmative studies are needed.


Pituitary tumor transforming gene (PTTG) and fibroblast growth factor (FGF)


PTTG is involved in regulatory functions determining control of many fundamental cellular events including mitosis, cell transformation, DNA repair and gene regulation. Several of these events are mediated through interactions with PTTG binding factor (PBF) and fibroblast growth factor-2 (FGF-2). McCabe et al. [69] showed that PTTG, FGF-2 and its receptor FGF-R-1 were all overexpressed in adenomas compared with normal pituitary tissue. Moreover, the significantly enhanced expression of FGF-R-1 was observed in invasive adenomas compared with other pituitary tumors. More recently, a meta-analysis comprising 752 patients with a pituitary adenoma further corroborated the relationship between PTTG and the invasiveness of pituitary adenomas [70].


Matrix metalloproteinases family


Matrix metalloproteinases (MMPs), A Disintegrin and Metalloproteinase (ADAM) and A Disintegrin and Metalloproteinase with Thrombospondin Motif (ADAMTS) are zinc-dependent endopeptidases that play a critical role in the destruction of extracellular matrix proteins related to proliferation and tumor invasion of various types [71,72]. Specifically in the pituitary, the level of these MMPs assessed by immunohistochemistry and/or western blot has been found to correlate with pituitary adenoma cell migration [73] and cavernous sinuous invasion [74,75]. However, other authors did not find this relationship [76,77].

## 3. Discussion

We reviewed the main underlying predisposing factors and molecular mechanism of pituitary apoplexy. The knowledge of the peculiarities of pituitary vascularization, development and the intrinsic blood supply of the pituitary adenoma is relevant to understanding the apoplectic event.

Pituitary apoplexy in the setting of a tumor seems to be favored by a reduced intra-tumor vascularization with inherent vascular fragility along with a rapid tumor growth associated with high metabolic requirements and increased intra-tumor pressure. All these features enhance the susceptibility for developing ischemia/infarction or bleeding, either in large or small tumors. Although the molecular mechanisms are yet to be elucidated, the reduced intra-tumor microvascular density could facilitate hypoxia mediated triggering of HIF-1α/VEGF pathways which, in combination with extracellular matrix remodeling due to the activation of MMPs, results in tumor growth and angiogenesis, thus making the pituitary tumor prone to apoplectic events.

Although we have performed an exhaustive review, the main limiting factor has been the scarce literature focused specifically on the study of molecular mechanisms specifically in the pathogenesis of pituitary apoplexy. In fact, the purpose of many of the selected papers addressed the study of aggressive pituitary tumors or the cancer development rather than to identify the mechanisms involved in pituitary apoplexy.

## 4. Conclusions and Remarks

Pituitary apoplexy is a life-threatening condition consisting of an intra-pituitary hemorrhage and/or infarction that in the vast majority of cases occurs in the setting of a pituitary tumor. A diligent clinical suspicion is essential because early treatment with intravenous corticosteroids is mandatory. Apart from several reported risk factors, the peculiarity of the blood supply of both the pituitary gland and the pituitary adenoma are important to understand the pathophysiology of the pituitary apoplexy. In this regard, it should be emphasized that the low vascularization of pituitary adenomas is crucial for triggering some molecular mediators involved in tumor hemorrhage. However, the underlying molecular pathways are yet to be fully elucidated, and we do not have any molecular marker to help physicians in clinical decision making

A multidisciplinary task force comprising basic and clinical researchers is needed to gain more insights into the underlying molecular mechanisms involved in pituitary apoplexy. This will permit us to design tools aimed at identifying tumors that may be more prone to bleeding, and once the apoplexy has occurred would even help us in optimizing its management.

## Figures and Tables

**Figure 1 ijms-23-08721-f001:**
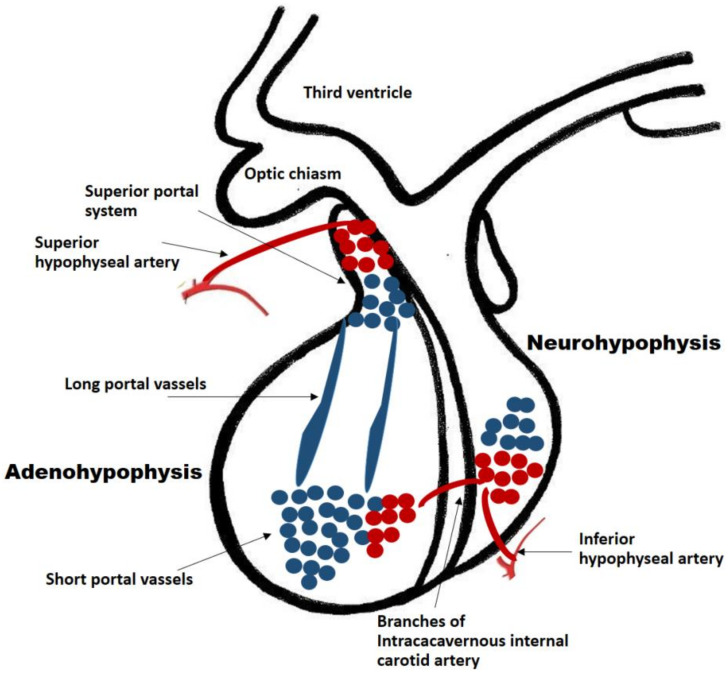
Pituitary blood supply and portal system. The adenohypophysis and neuro-hypophysis receive their blood supply primarily from the internal carotid arteries. The infundibulum, the median eminence and the pars tuberalis are supplied by the superior hypophyseal artery (a branch of the internal carotid artery). The posterior pituitary receives its blood supply directly from the inferior hypophyseal artery. The anterior pituitary is supplied indirectly from two sources: (a) Long portal vessels which supply 70% of the blood to the gland and arise from the superior hypophyseal artery; (b) Short portal vessels which supply 30% of blood to the gland and arise from the posterior pituitary. These portal vessels are exquisitely sensitive to volume and pressure changes in the systemic circulation.

**Figure 2 ijms-23-08721-f002:**
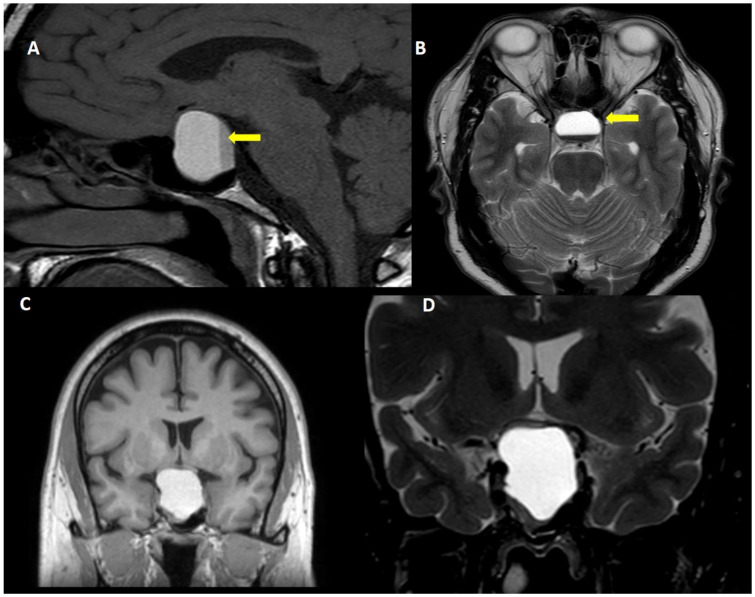
Acute pituitary apoplexy. A 40 year-old man, attended to in the emergency room due to sudden onset headache, vomiting and visual symptoms. (**A**): T1W sagittal view, (**B**): Fluid attenuated inversion recovery view, (**C**): T1W coronal view and (**D**): T2W coronal view. Note the fluid-fluid level within the tumor (yellow arrow).

**Figure 3 ijms-23-08721-f003:**
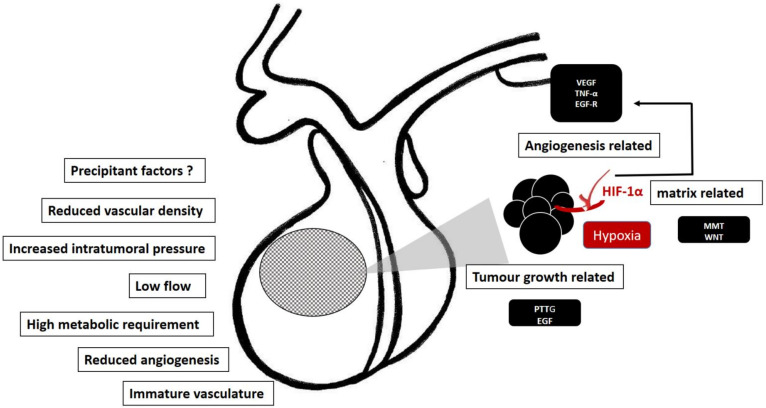
Predisposing factors and molecular mechanism involved in the pathogenesis of pituitary apoplexy. Some intrinsic characteristics facilitating the apoplectic event (left side). Hypoxia is a plausible molecular trigger of hypoxia-mediated factors, HIF-1α/VEGF and MMP pathways, promoting the tumor growth and angiogenesis, thus making the pituitary tumor prone to apoplectic events (right side).

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
