# Peer review of "Pituitary Apoplexy: Risk Factors and Underlying Molecular Mechanisms"

_ijms, 2022, doi:10.3390/ijms23158721_

Round 1

Reviewer 1 Report

The ms by Bigetti & Simo touches very important condition of pituitary apoplexy. However, being rare disease, it's need to be diagnosed and treated. The ms has large potential, but, in my opinion, it needs some corrections / explanations:

1. The ms needs overall thorough reading and correcting many punctuation and grammatical mistakes, most of them being the rest from the template (see Title).

2. Title needs some punctuation (hyphen?).

3. The ms describes very well the pathogenesis of pituitary apoplexy. But the comparison with brain stroke and pituitary peculiarities (as the title says) is not well grounded and explained. The brain stroke itself is clinically rather different from pituitary apoplexy, so the inclusion of that condition here is rather weird. I would recommend either reformulating the title and focusing on the pituitary or better explaining the concept. Moreover, if comparing, a table would be of value.

4. The ms definitely lacks a description of the clinical symptoms of pituitary apoplexy and the treatment. Some MRI pictures? If included - also stroke symptoms.

5. Conclusions' section

- "Due to the different embryogenesis and vasculogenesis, the apoplectic event in the pituitary gland is totally different to a CNS stroke." - I am very sorry to write that, but the difference is obvious for every clinician. Different symptomatology, simply. That's why putting those two conditions together should be better grounded.

- "There is an urgent need to create a multidisciplinary task force" - always there's a need, but urgent? Rare disease, it's not epidemiological problem.

- for conclusion, except of formating the team, 1-2 real conclusions, like potential markers, differentiating symptoms, should be mentioned.

Author Response

The ms by Bigetti & Simo touches very important condition of pituitary apoplexy. However, being rare disease, it's need to be diagnosed and treated. The ms has large potential, but, in my opinion, it needs some corrections / explanations:

Many thanks for the revision process and your constructive criticism, which has been essential to improve the quality of our paper.

  1. The ms needs overall thorough reading and correcting many punctuation and grammatical mistakes, most of them being the rest from the template (see Title).

Answer: We have extensively corrected all the punctuation and grammatical mistakes.

  1. Title needs some punctuation (hyphen?).

Answer: Title has been significantly changed.   

  1. The ms describes very well the pathogenesis of pituitary apoplexy. But the comparison with brain stroke and pituitary peculiarities (as the title says) is not well grounded and explained. The brain stroke itself is clinically rather different from pituitary apoplexy, so the inclusion of that condition here is rather weird. I would recommend either reformulating the title and focusing on the pituitary or better explaining the concept. Moreover, if comparing, a table would be of value.

Answer: Following the reviewer recommendation, we have leave out the comparison with the brain stroke and we have focused on pituitary apoplexy. In addition, the title has been reformulated accordingly.

  1. The ms definitely lacks a description of the clinical symptoms of pituitary apoplexy and the treatment. Some MRI pictures? If included - also stroke symptoms.

Answer: Thanks for mention it! We have added a new section focused in pituitary apoplexy presentation and treatment. Page 7 lines 258-273. We have added an MRI image as recommended.  Many thanks for your suggestion!!

  1. Conclusions' section

- "Due to the different embryogenesis and vasculogenesis, the apoplectic event in the pituitary gland is totally different to a CNS stroke." - I am very sorry to write that, but the difference is obvious for every clinician. Different symptomatology, simply. That's why putting those two conditions together should be better grounded

Answer:  As recommended we have addressed  the review focusing only in pituitary apoplexy and, consequently, this  sentence has been deleted.

- "There is an urgent need to create a multidisciplinary task force" - always there's a need, but urgent? Rare disease, it's not epidemiological problem.

Answer:  We tone down this expression as followed: “A multidisciplinary task force comprising basic and clinical researchers is needed.. ”  page 12 lines 496-497

- for conclusion, except of formating the team, 1-2 real conclusions, like potential markers, differentiating symptoms, should be mentioned.

Answer:  We have added a couple of conclusions in the revised manuscript (line 8 pages 586-501

Reviewer 2 Report

The authors deal with a relevant and probably underestimated topic, i.e., the mechanisms implicated in pituitary apoplexy based on hints from stroke and pituitary peculiarities. To this end, they narratively reviewed the literature comparing pituitary apoplexy and cerebral stroke in order to highlight the differences in terms of ontogenesis, vascular system, and apoplexy mechanisms themselves. The authors also provided an overview of the potential pituitary adenoma susceptibility for apoplectic events and molecular pathways that could be involved, with the final aim to summarize the state of the art and to gain deeper insight into this rare but sometimes fatal condition. Overall, the review is nicely conceived, the studies included are consistent and are adequately illustrated. Few comments requiring some revision.

Title: please remove the first word (“Title”) and rephrase the sentence “… pituitary apoplexy lesson from brain stroke and …” into the following: “… pituitary apoplexy: lessons from stroke and …”

Abstract: please provide more details on the results reviewed and briefly include the translational implications of these findings and their potential clinical applications in the diagnosis and management.

Introduction: please briefly describe clinical presentations of these conditions, including those due to non-functioning giant pituitary adenomas, as recently reported and reviewed (i.e., PMID: 35329907). Please also expand the final part of this section by clearly stating the aim and novelty of the present review.

Section 2 and 3: although interesting, these sections may be shortened and more focused on the main target of the present review.

Methods/Results: although the narrative design of this review, a brief “Methods” section describing the search strategy, inclusion/exclusion criteria, and selection procedures should be mentioned. The same holds true for a short “Results” section showing the number of studies retrieved, selected, and eventually included. Alternatively, the authors may succintly state how the studies here reviewed have been selected.

Table: a table summarizing the main methodology and findings of the studies reviewed would be helpful.

Discussion: before “Conclusions and remarks”, please include a short “Discussion” section briefly summarizing the main findings reviewed here and their translational applications/clinical implications. At the end of this new section, please also include the limitations of the studies reviewed, any literature gap, and future applications/research agenda.

References: please check the list for style, accuracy, and completeness.

General: although the language is overall acceptable, an editing by a native-English speaker would be useful.

Author Response

The authors deal with a relevant and probably underestimated topic, i.e., the mechanisms implicated in pituitary apoplexy based on hints from stroke and pituitary peculiarities. To this end, they narratively reviewed the literature comparing pituitary apoplexy and cerebral stroke in order to highlight the differences in terms of ontogenesis, vascular system, and apoplexy mechanisms themselves. The authors also provided an overview of the potential pituitary adenoma susceptibility for apoplectic events and molecular pathways that could be involved, with the final aim to summarize the state of the art and to gain deeper insight into this rare but sometimes fatal condition. Overall, the review is nicely conceived, the studies included are consistent and are adequately illustrated. Few comments requiring some revision.

Many thanks for the revision process and your kind comments on our manuscript

Title: please remove the first word (“Title”) and rephrase the sentence “… pituitary apoplexy lesson from brain stroke and …” into the following: “… pituitary apoplexy: lessons from stroke and …”

Answer: The tile has been reformulated because R1 strongly recommend to leave out the comparison with stroke.   

Abstract: please provide more details on the results reviewed and briefly include the translational implications of these findings and their potential clinical applications in the diagnosis and management.

Answer: The abstract has been modified accordingly. Thanks for mention it! (Page 1, lines 13-23.

Introduction: please briefly describe clinical presentations of these conditions, including those due to non-functioning giant pituitary adenomas, as recently reported and reviewed (i.e., PMID: 35329907). Please also expand the final part of this section by clearly stating the aim and novelty of the present review.

Answer: We have added a sentence with the clinical presentation. Page 1, lines 31-32 and a phrase regarding the susceptibility of NFPAs to apoplexy including in the giant ones with the new reference. Page 1 lines 41-44.  New reference number 8  is highlighted in yellow. As recommended, a clear statement on the aim and novelty of the review has been added to the end of this section.

Section 2 and 3: although interesting, these sections may be shortened and more focused on the main target of the present review.

Answer: We agree with the referee and, therefore, we have deleted section 3 and the  section of pituitary development has been significantly reduced (page 2 lines 81-87).

Methods/Results: although the narrative design of this review, a brief “Methods” section describing the search strategy, inclusion/exclusion criteria, and selection procedures should be mentioned. The same holds true for a short “Results” section showing the number of studies retrieved, selected, and eventually included. Alternatively, the authors may succintly state how the studies here reviewed have been selected.

Table: a table summarizing the main methodology and findings of the studies reviewed would be helpful.

Answer: We have added a new section called methods explaining the search strategy (page 2 lines 58-68).

Discussion: before “Conclusions and remarks”, please include a short “Discussion” section briefly summarizing the main findings reviewed here and their translational applications/clinical implications. At the end of this new section, please also include the limitations of the studies reviewed, any literature gap, and future applications/research agenda.

Answer:  Following your recommendation, we have added a short “Discussion” section, comprising all the items suggested by the reviewer (page: 12 lines: 465-483).

References: please check the list for style, accuracy, and completeness.

Answer: Done! New references have been highlighted in yellow

General: although the language is overall acceptable, an editing by a native-English speaker would be useful.

Answer: English has been reviewed throughout the manuscript.   

Round 2

Reviewer 1 Report

The authors satisfactorily improved the paper.

I have no other comments.

Author Response

Many thanks for the revision process, which has improved the quality of our manuscript